# Direct and Indirect Genotoxicity of Graphene Family Nanomaterials on DNA—A Review

**DOI:** 10.3390/nano11112889

**Published:** 2021-10-28

**Authors:** Kangying Wu, Qixing Zhou, Shaohu Ouyang

**Affiliations:** 1Key Laboratory of Pollution Processes and Environmental Criteria (Ministry of Education)/Tianjin Key Laboratory of Environmental Remediation and Pollution Control, Nankai University, Tianjin 300350, China; wukangyingcool@163.com; 2College of Environmental Science and Engineering, Nankai University, Tianjin 300350, China

**Keywords:** graphene family nanomaterials, genotoxicity, DNA damage, safety, toxicity

## Abstract

Graphene family nanomaterials (GFNs), including graphene, graphene oxide (GO), reduced graphene oxide (rGO), and graphene quantum dots (GQDs), have manifold potential applications, leading to the possibility of their release into environments and the exposure to humans and other organisms. However, the genotoxicity of GFNs on DNA remains largely unknown. In this review, we highlight the interactions between DNA and GFNs and summarize the mechanisms of genotoxicity induced by GFNs. Generally, the genotoxicity can be sub-classified into direct genotoxicity and indirect genotoxicity. The direct genotoxicity (e.g., direct physical nucleus and DNA damage) and indirect genotoxicity mechanisms (e.g., physical destruction, oxidative stress, epigenetic toxicity, and DNA replication) of GFNs were summarized in the manuscript, respectively. Moreover, the influences factors, such as physicochemical properties, exposure dose, and time, on the genotoxicity of GFNs are also briefly discussed. Given the important role of genotoxicity in GFNs exposure risk assessment, future research should be conducted on the following: (1) developing reliable testing methods; (2) elucidating the response mechanisms associated with genotoxicity in depth; and (3) enriching the evaluation database regarding the type of GFNs, applied dosages, and exposure times.

## 1. Introduction

Graphene, a two-dimensional crystal repeatedly peeled from graphite, is a single layer of carbon atoms with a sp^2^-hybridized structure (Figure 1a) [1,2]. Graphene and its derivatives, including graphene oxide (GO), reduced graphene oxide (rGO), and graphene quantum dots (GQDs), exhibit various excellent physical, electrochemical, and optical advantages [3,4,5]. GO is an amphiphilic sheet-like graphenic carbon and contains fewer oxygen functional groups (Figure 1b) [6,7]. rGO is prepared by oxidative exfoliation of graphite and has lower C/O ratios than GO (Figure 1c) [8]. GQDs are similar to graphene but have unique zero-dimensional structures due to their nanoscale sized lateral dimensions (Figure 1d) [9].

Currently, GFNs, as promising nanomaterials, have attracted increasing attention in the scientific community and are in commercial production for many applications, such as energy storage [10,11,12,13,14,15,16,17], medicine [18,19,20,21,22,23,24,25], environmental protection [26,27,28,29,30,31], and industrial manufacturing [32,33,34]. For example, the market for graphene-based products is forecast to reach $675 million by 2020 [35]. With rapid developments in application and production of GFNs, their potential for release into the environment and the environmental risks of GFNs have become emerging issues [36,37,38]. Consequently, many studies have shown that adverse effects can be induced by GFNs in vivo and in vitro, such as organ (e.g., lung, liver, and spleen) toxicity, cytotoxicity, immunotoxicity, neurotoxicity, and reproductive and developmental toxicity [3,39]. Moreover, the toxicity mechanisms of GFNs to organisms, including physical destruction, oxidative stress, inflammatory response, apoptosis, autophagy, and necrosis, are summarized in Table 1. However, the genotoxicity of GFNs on DNA (e.g., DNA damage) remains largely unknown.

Genotoxicity is broadly defined as ‘damage to the genome’ and also a distinct and important type of toxicity, as specific genotoxic events are considered hallmarks of cancer [40]. Generally, the genotoxicity can be sub-classified into direct genotoxicity and indirect genotoxicity in cells or the nucleus [41,42,43]. Nanoparticles (NPs) can be uptaken by the nucleus and induce DNA damage, leading to direct genotoxicity on organisms [42]. While many studies have shown that most NPs cannot enter the nucleus, they still indirectly affect genotoxicity by oxidative stress, epigenetic changes, inflammation, and autophagy [42]. Moreover, genotoxicity plays a key role in assessing the safety of NPs on human health and the environment [44,45,46,47]. Although there has been many researches about the genotoxicity of NPs in recent years, it is mainly focused on traditional artificial nanomaterials, such as TiO_2_, carbon nanotubes, and silver and gold NPs [48,49,50]. However, the existing literature on genotoxicity of GFNs remains limited and conflicting. A few studies showed that GFNs had no adverse effects on genotoxicity [51]. In contrast, many researchers have reported that the small size and sharp edges of GFNs (e.g., GO and GQDs) can induce genotoxicity on aquatic organisms (e.g., fish and algae) [52,53,54]. However, the direct and indirect genotoxicity mechanisms of GFNs remain unclear, despite genotoxic phenomena being widely reported.

The purpose of this article is to critically review the existing literatures on the genotoxicity of GFNs. This review will focus mainly on the genotoxicity mechanisms of GFNs in order to (1) expand our understanding of possible mechanisms underlying the promotion of DNA damage by GFNs; (2) highlight the direct and indirect genotoxicity of different subsets of GFNs; and (3) explore the factors that influence the genotoxicity of GFNs. This review will provide new insights into the genotoxicity and environmental risks of engineered nanoparticles (ENPs).

**Table 1 nanomaterials-11-02889-t001:** The toxicity of GFNs in vivo and in vitro.

Products	Supplier or Synthesis Methods	Dose	Animal or Cell Models	Toxicological Mechanisms	Adverse Effects	Ref.
graphene nanoplatelets	cheaptubes.com (Brattleboro, VT, USA)	0.3, 1 mg/rat	rat	oxidative stress, inflammation	lung inflammation	[55]
commercial GO and rGO	Nanjing XFNANO Materials Tech Co., Ltd., (China)	2.0 mg/kg body weight	rat	transcriptional and epigenetic	liver zonated accumulation	[56]
amination GQDs carboxylated GQDs hydroxylated GQDs	Nanjing XFNANO Materials Tech Co., Ltd., (China)	100, 200 μg/mL	A549 cells	autophagy	cytotoxicity	[57]
GO and rGO oxidated from carbon nanofibers	Grupo Antolin (Spain)	0.1, 1.0, 10, 50 mg/L	erythrocyte cell	oxidative stress	genotoxicity	[58]
GO nanosheets	Sigma-Aldrich (St. Louis, MO, USA)	40, 60, 80 mg/L	Human SH-SY5Y neuroblastoma cell	oxidative stress, autophagy–lysosomal network dysfunction	cytotoxicity	[59]
pristine rGO	Chengdu Organic Chemicals Co., Ltd., the Chinese Academy of Sciences	1–100 mg/L	Earthworm coelomocytes	oxidative stress	immunotoxicity	[60]
single layer GO (product no. GNOP10A5)	ACS Materials LLC (Medford, MA, USA)	1, 10, 50, 150, 250, 500 mg/L	*Escherichia coli*	physical destruction	toxicity against bacteria	[61]
GO	modified Hummers method	25 mg/L	THP-1 and BEAS-2B cells	lipid peroxidation, membrane adsorption, membrane damage	cytotoxicity	[62]
GO	modified Hummers method	2 mg/kg	rat	lipid peroxidation, membrane adsorption, membrane damage	acute lung inflammation	[62]
GO	Nanjing XFNANO Materials Tech Co., Ltd., (China)	0–100 mg/L	zebrafish embryos	oxidative stress	developmental toxicity	[63]
GO	modified Hummers method	10 mg/L	*Caenorhabditis elegans*	oxidative stress	toxicity	[64]
graphene, GO	modified Hummers method	3.125–200 mg/L	human erythrocytes and skin fibroblasts	oxidative stress	cytotoxicity	[65]
graphene exfoliated form graphite, GO oxidated from carbon fibers	Grupo Antolin Ingeniería (Burgos, Spain)	1, 10 mg/L	primary neurons	inhibition of synaptic transmission, altered calcium homeostasis	neurotoxicity	[66]

## 2. Direct Genotoxicity of GFNs

Adsorption on DNA is a critical physiochemical process at the DNA–GFNs interface due to large surface area and surface-active properties of the GFNs. This process can (1) induce the direct genotoxicity of GFNs (e.g., DNA damage); and (2) alter the function of DNA through coating and modification by GFNs. In reviewing the current literature, the adsorption of GFNs on DNA and molecular interactions were investigated.

### 2.1. Direct Physical Nucleus Damage by GFNs

After GFNs exposure, monolayer or a few-layer GFNs (GO and rGO) sheets are able to cut and penetrate cell membranes and the cell wall (if present), resulting in direct physical membrane damage [67,68]. Moreover, small pieces of GFNs will enter the nucleus, interacting directly with DNA [69]. Generally, nuclear DNA is the main target of gene toxicity [42]. Prokaryotes (e.g., bacteria) only have naked DNA without a nuclear envelope. GFNs can directly contact bacteria RNA/DNA hydrogen groups, interrupting the replicative stage after internalization [70]. During mitosis, GFNs are likely to interact with DNA, leading to DNA aberration when the nuclear membrane ruptures [3]. As shown in Figure 2, the nuclear uptake and nuclear response related to contact with GQDs have been systematically reported by using atomic force microscopy (Figure 2a,b), confocal microscopy (Figure 2c,d), transmission electron microscopy (Figure 2e,f), and high content screening (Figure 2g,h) [69]. GQDs are mainly uptaken into cells via energy-dependent endocytosis, phagocytosis, and caveolae-mediated endocytosis. More than half of GQDs are exposed and accumulated in the nucleus by microscopy investigation. The accumulated GQDs may direct contact with DNA strand, thereby causing physical damage. After 1 h exposure, the rGO nanoplatelet can pierce the nucleus of the human mesenchymal stem cells (hMSCs), leading to DNA fragmentation and chromosomal aberrations at 0.1 and 1.0 mg/L. Notably, rGO sheets with the same size or larger size showed no genotoxicity in the hMSCs after 24 h exposure at 100 mg/L [71]. The single-layer rGO nanoribbons can penetrate into the hMSCs nucleus at 100 mg/L detected by confocal fluorescence imaging, and cells showed a high degree of DNA fragmentation. The above DNA damage is mainly related to oxidative stress caused by DNA released, rather than DNA damage within the nucleus. Interestingly, rGO nanoribbons showed no significant cytotoxicity at 1.0 mg/L but can induce genotoxicity through DNA fragmentation and chromosomal aberrations in the hMSCs [72]. In a word, GFNs can interact directly with chromatin and DNA, causing DNA damage and thus exhibiting genotoxicity.

### 2.2. Interaction Mechanisms between DNA and GFNs

Interaction between GFNs and DNA is critical for understanding and assessing direct genotoxicity. DNA is a biological macromolecule with a repetitive nucleotide structure that controls biological functions. The backbone of DNA is a regular sequence of deoxyribose sugar and phosphate groups. DNA is always negatively charged at most pH values [73]. The oxidized domains of GO are rich in oxygen-containing groups (e.g., epoxides, hydroxyl, carboxyl, and carbonyl groups). The negatively charged carboxyl groups would electrostatically repel negatively charged DNA and use hydrogen bonding as the main attraction force [47]. The GFNs mainly interact with DNA via H-bonding and π–π stacking, casing the DNA distortion and even DNA cleavage. The DNA damaging mechanism of GQDs depends on their size. The small GQDs easily enter the DNA molecule leading to DNA base mismatch. Large GQDs tend to stick to the ends of the DNA molecule, causing the DNA to unfold [74]. The zipper-like unfolding of double-stranded DNA caused by graphene wrinkles has been investigated by using molecular dynamics simulations. The results show that the zipper pattern brings more DNA bases into contact with the wrinkled region, resulting in accelerated deformation of double-stranded DNA [75]. The GO combining with copper ions can intercalate into DNA molecules and cleave DNA fragments, and the system of this DNA cleavage is oxidative and hydrolytic [76]. Unlike AuNPs, which rely on stronger DNA base coordination, the adsorption of GO is weak, owing to the weak binding affinity [77]. Furthermore, the GO surface shows great heterogeneity for DNA adsorption and hydrophobic regions for exclusion of DNA. Thus, both the external environment and the physicochemical property (e.g., oxidized degree and size) have a strong influence on the adsorption capacity of GO [47,76,77,78]. Further research into the direct effects of GFNs on DNA or genetic material is important to explain GFN-mediated targeted genotoxicity.

## 3. Indirect Genotoxicity of GFNs

Although GFNs can induce direct genotoxicity, most of the current studies focus on GFNs’ indirect genotoxicity on the indirect effect on gene normal tissue expression. Indirect genotoxicity covers different aspects. Here, we describe the indirect genotoxicity of GFNs in the following aspects: oxidative stress, epigenetic toxicity, DNA replication, repair and transcription affected by GFNs, and inflammation and autophagy.

### 3.1. Oxidative Stress

The internalization NPs by organism can induce intracellular reactive oxygen species (ROS) generation and antioxidant defense. ROS generation can lead to typical oxidative DNA damage (e.g., single- and double-stranded DNA breaks, DNA cross-links, and base modifications) [78,79,80]. Indirect genotoxicity of GFNs mediated by oxidative stress has been explored in vivo and in vitro. For instance, ROS generation and ROS-dependent DNA damage and genotoxicity were observed in human retinal pigment epithelium (ARPE-19) cells after 24 h exposure to GO and rGO [81]. Similarly, GO and rGO can also trigger genotoxicity of female C57BL/6J mice by induction of oxidative stress [82]. Exposed to few-layer graphene (FLG), the indirect DNA damage in THP-1 macrophages and human-transformed type-I alveolar epithelial cells was also driven by oxidative stress [43]. The specific induced mechanisms of indirect DNA damage are identified by baseline levels of micronuclei induction. Moreover, the indirect genotoxicity induced by FLG also correlates with an increase of inflammatory mediator (IL-8), decreased antioxidant (rGSH), and a depletion in mitochondrial ATP production [83]. Zhao et al. reported that GO can induce oxidative stress and genotoxicity in earthworms and the excessive accumulation of ROS, leading to lipid peroxidation, lysosomal membrane damage, and DNA damage [84]. Organisms possess a well-developed inhibition of antioxidant defense, including ROS-scavenging enzymes (e.g., superoxide dismutase (SOD), peroxidase, and catalase) and regulatory mechanisms to protect organisms from the negative effects of ROS [46,84]. The ROS generation benefitted from inhibition of fatty acid, carbohydrate, and amino acid metabolism [85]. ROS induced by GO seemed to be the main mechanism leading to human lung fibroblast (HLF) cells of genotoxicity [86]. Natural nanocolloids (Ncs) can mediate the phytotoxicity of GO such that GO–Ncs induced stronger ROS production and DNA damage compared with GO alone [87]. The mitochondrial oxidative stress induced by GQDs in microglia can cause ferroptosis.

### 3.2. Epigenetic Toxicity

Epigenetic regulatory mechanisms can be observed after exposure to NPs, including DNA methylation, histone modification, non-coding RNA (ncRNA) gene expression regulation, and dynamic chromatin organization [88,89]. As a response to internal and external stimuli, these above epigenetic regulations and complex, time-specific, and tissue-specific control of gene expression were allowed during development and differentiation [90]. DNA methylation, a covalent modification of cytosine residues in DNA, plays a supreme role in the stabilization and regulation of gene expression during development or differentiation [91,92]. Ting et al. [91] firstly proved that GQDs can inhibit the DNA methylation of transcription factor Sox2 and regulated DNA methyltransferase and demethyltransferase expressions. Global DNA hypomethylation of caprine fetal fibroblast cells, which are exposed to GO-AgNPs, might result from oxidative stress [93]. Histone modifications containing phosphorylation, methylation, and acetylation also are major components of epigenetic regulatory mechanisms [92]. The role of epigenetic regulation about toxicity of GFNs has been described in human embryonic kidney 293T cells [89]. The results showed that the GO triggered the formation of new intra-chromosomal looping (A1–A3) and enhanced and promoted cyclo-oxygenase-2 (Cox2) expression and activation. The epigenetic mechanisms of GO on transgenerational reproductive toxicity were determined using a house crickets generational experiment [94].

GO can activate microRNA (miRNA) protection regulation and inhibit the reproductive toxicity of *Caenorhabditis elegans*, which was also an epigenetic signal encoded protection mechanism [95]. Moreover, miRNAs can activate death receptor pathways by altering the expression of caspase-3 and tumor necrosis factor α receptor in GO-exposed pulmonary adenocarcinoma (GLC-82) cells [96]. Therefore, the epigenetic process induced by GFNs are complex and multi-layered. Currently, the existing studies are mainly limited to the reactions of epigenetic toxicity induced indirect genotoxicity of GFNs. How to explain the causal epigenetic mechanisms induced by GFNs remains challenging. Future experimental studies should be carefully designed for better understanding the genotoxic effects of GFNs induced epigenetic modifications that directly or indirectly cause DNA damage.

### 3.3. The DNA Replication, Repair, and Transcription Affected by GFNs

GFNs have the ability to alter gene expression by interacting with signal transduction cascades or replication/repair/transcription mechanisms [97,98]. GO exposure activates a variety of signaling pathways, triggering the expression of many kinds of genes related to autophagy, apoptosis, and necrosis [89,99]. Cell apoptosis and the upregulation of the tumor protein p53 gene in the cell cycle induced by both nano- and microsized GO was detected [99]. In the work, both nano- and microsized GO block the cell cycle in the S phase, a critical period in the cell cycle. The GQDs (100 mg/L) can induce genotoxicity through ROS generation and inhibition of gene regulation in the cell cycle of rat alveolar macrophage cells [100]. The key genes (such as RAD51, BRCA2, ATM, and PARP1) regulate some key biological processes (e.g., nucleosome assembly, stress response, protein folding, and DNA damage) in FLG-exposed human primary endothelial cells [97]. Moreover, related study have shown that GFNs may cause genotoxicity by affecting the nucleotide excision repair and the repair system of non-homologous end connections [101].

### 3.4. Inflammation

Inflammation, including acute and chronic inflammation, is a complex biological response to harmful stimuli such as pathogens, poisons, or dead cells [102]. GO induced high expression of Cox2, a hallmark of inflammation and which is involved in acute and chronic diseases [103]. Inflammation is also one of the reactions of ROS induced indirect genotoxicity [104]. Chronic inflammation can induce secondary genotoxicity, which is manifested in the accumulation of reactive oxygen species, after GFNs exposed to cells [43,105]. Interestingly, there was no oxidative damage and a weak anti-inflammatory response for assessing the potential genotoxicity of GO and graphene nanoplatelets in the human intestinal barrier in vitro model simulation [106]. However, both GO and GNPs can induce DNA breaks, and GO can activate the nuclear factor kappa-B signaling pathway, which may lead to macrophage inflammation [107]. Excess inflammatory cytokines can cause DNA damage [108]. There are complex causal interactions between inflammation and ROS, and they may have independent induction mechanisms. In summary, the genotoxicity of GFNs mediated by inflammation can be attributed to the direct stimulation, secondary effect of cytokine release or ROS accumulation.

### 3.5. Autophagy

Autophagy, a cell survival mechanism, is described as a highly regulated intracellular catabolic pathway involving degradation of unnecessary or dysfunctional components to maintain cell homeostasis [109,110]. Autophagy controls transformation of nuclear components (e.g., nuclear lamina, chromatin, and DNA), which is important for maintaining genomic stability [111]. Inhibition of autophagy obstructs normal DNA damage repair and induces cell death in response to genotoxic stress. GFNs can induced ROS generation in mitochondria, which begin to exert autophagy to avoid oxidative damage and to reduce mutation of mitochondrial DNA [112]. GO was able to result in accumulation of autophagosomes, reduction in autophagic degradation, and lysosomal impairment [113]. Autophagy and epigenetic changes jointly regulate cell survival, and autophagy may be a downstream mechanism of epigenetic changes, one of the manifestations of secondary genotoxicity [114]. Graphene oxide quantum dot exposure induced autophagy in a ROS-dependent manner [115]. The relationship between autophagy and DNA damage is complex, while autophagy can regulate the levels of various proteins participating in the repair and detection of damaged DNA [116]. The relationship between autophagy and other toxicity mechanisms (e.g., oxidative stress, epigenetic changes, apoptosis, and inflammation) of other GFNs is still unclear [114]. Understanding GFNs-mediated autophagy is of great significance to explain the genotoxicity of GFNs.

## 4. Factors Influencing Genotoxicity of GFNs

As is known to all, there is a strong correlation between cytotoxicity and the physicochemical properties of NPs, such as particle size and shape, surface characteristics, and surface functionalization. Similarly, the genotoxicity of GFNs can be affected by these factors [117]. The genotoxicity of GFNs is greatly varied in the literature, which can be attributed to numerous factors including physicochemical properties (morphology, surface chemistry, size, shape, and purity), dose, test species, exposure time, and exposure assay [80,118].

### 4.1. Surface Properties

The oxygen-containing functional groups play a key role in the genotoxicity of GFNs [58,81,82,83,119]. For example, the rGO with lower oxygen content can induce stronger genotoxicity on ARPE-19 cells than these GO with higher oxygen content, suggesting that GO has a better biocompatibility owing to more saturated C–O bonds [81]. The remove of epoxy groups from the GO surface mitigates GO in vivo genotoxicity toward *Xenopus laevis* tadpoles [58]. Compared with GO, graphene, rGO, and graphite all induce higher levels of genotoxicity in glioblastoma multiforme cells, and the difference was attributed to the hydrophilic and hydrophobic surface and edge structure of GFNs [119]. GO has hydrophilic properties and smooth and regular edges, while rGO and graphene have hydrophobic properties and sharp and irregular edges, which can damage the integrity of cell membranes greatly. The carboxyl groups in the surface of carboxyl-FLG may scavenge oxidative radical on bronchial epithelial cells to alleviate the genotoxicity of FLG [83]. Moreover, different immunological mechanisms triggered by GFNs can be attributed to the proportion of hydroxyl groups [82]. Cells produce a stronger inflammatory response after being exposed to GO than rGO by detecting transcriptomic changes, and the reason is attributed to the large number of hydroxyl groups on the surface of GO [82]. The surface functionalization also can significantly modulate the toxicity of GFNs [53,85,86,120]. For example, amino functionalized GQDs induced lower ferroptosis effects than nitrogen-doped GQDs [85]. Similarly, the DNA methylation of various tissues induced by GQDs was depend on their different surface chemical modifications [53]. Increased cytotoxicity and genotoxicity of the aminated GO were detected by following 24 h exposure on Colon 26 cells [120]. A study on the genotoxicity reduced by GO and rGO showed that the GTPs-rGO reduced by green tea polyphenols (GTPs) yielded more biocompatible and reduced sheets with lower genotoxic effects, as compared to the N_2_H_4_–rGO, which were reduced by hydrazine (N_2_H_4_) [121]. The acid-polyethylene glycol (LA-PEG) and PEG modified GO induced gentle DNA damage and decreased the genotoxicity of GO to HLF cells [86]. Surface charge also influences significantly the genotoxicity of GFNs [86,122]. The genotoxic effect of GO on cells is proportional to the amount of positive charge on the surface [86]. The surface charge density of graphene in aqueous solution can transform to chemically-converted graphene, leading to the capture of large amounts of DNA [122]. The different hydrophilic and hydrophobic properties of GO/rGO regulated by differential surface chemistry (especially the O/C ratio) determine the potential of graphene to interact with organisms [123,124,125]. Despite hydrophilic and hydrophobic rGO exhibiting similar toxic responses (e.g., cytotoxicity, DNA damage, and oxidative stress) to cells, their biological and molecular mechanisms are different [123]. The hydrophilic GO and hydrophobic rGO induce both kinds of DNA damage, namely single stranded and double stranded breaks, but the dose dependency was very significant and evident in GO exposure in DNA damage but not in rGO exposure [123]. Hydrophilicity, also an important factor in determining the biocompatibility and colloidal stability of GFNs, leads to different interactions with cells and bio-distribution of GFNs [124,125]. For example, simple accumulation of hydrophobic pristine graphene on the surface of monkey kidney cells without any cellular internalization led to severe metabolic toxicity, whereas hydrophilic GO was internalized by the cells and concentrated near the perinuclear region without causing any toxicity under lower concentrations [124]. Therefore, the surface properties play an important role in understanding the genotoxicity manifestations and biological and molecular mechanisms of GFNs.

### 4.2. Size and Structure

The genotoxicity of GFNs within organisms is size-dependent. Compared with large GFNs, small GFNs have bigger surface areas and provide more sites to interact with cells, leading to greater cellular uptake of GFNs [126]. The size effect plays a key role in the genotoxicity of GFNs. For example, small rGO (average lateral dimensions 114 nm) induce higher genotoxicity in the hMSCs than large rGO (3.8 ± 0.4 μm) at 0.1 and 1.0 μg/mL after 1 h exposure. The lateral size and extremely sharp edged structure of GFNs can result in higher permeability to the cell and nucleus, resulting in greater genotoxicity. Similarly, the size of GFNs is an important determinant of subcellular penetration [126]. Li et al. [127] suggested that the larger the lateral size of GO, the more severe is the pyroptosis induced by GO in Kupffer cells. Moreover, there is a strong correlation between the size of GO and the structural change in small-interfering RNAs [128]. The large GO merely reduces the A-helical pitch, while small GO inserted into the double strands can wreak havoc on the RNA conformation [129]. In addition, Kong et al. [74] proved that the DNA damage mechanism of GQDs was limited by the size of GQDs through molecular dynamics simulations. Briefly, the relatively large GQDs (61 benzene rings) tend to stick to the ends of the DNA molecule, causing the DNA to unfold, while the small GQDs (seven benzene rings) are easily embedded in DNA molecules, leading to DNA base mismatches. The planar structure of GFNs may also have an effect on DNA damage. The dsDNA bases have a stronger binding affinity with wrinkled GFNs and even cause more DNA damage than with planar GFNs [75]. Given these discordant results, it is necessary to clarify the size- and structure-related genotoxicity of GFNs.

### 4.3. Exposure Dose and Time

The dose–response relationship is an important principle in nanotoxicology [42]. The modified GQDs may induce DNA hypermethylation in a time and dose dependent manner [53]. The high-dose (50 mg/L) GO induces more serious DNA methylation (hypermethylation) than low-dose (10 mg/L) treatment [101]. The effective accumulation of GFNs in the nucleus is regulated by two nuclear pore complex genes (Kapβ2 and Nup98), and their cellular internalization and absorption are related to exposure time [69]. Notably, the rGO sheets with the same size or larger size, higher concentration (100 μg/mL), and longer exposure time (24 h) showed no obvious genotoxicity in the hMSCs [71]. Overall, there are few studies on the genotoxicity of GFNs doses, and especially the combination of GFNs type and dose exposure is rare.

### 4.4. The Resistance of Cell Structures and Biological Barriers

From an organism’s perspective, the responses of various types of cells, organs, and tissues with different structures and functions to GFNs exposure were highly diverse. Internalization and direct contact membrane stress with extremely sharp edges of GFNs are considered as important mechanisms of toxicity [130,131]. For different bacterial models to graphene toxicity, the outer membranes can better “protect” bacteria from graphene [132]. The biological barrier is crucial for mammals against the damage from GFNs [3,117]. Both GO and graphene were able to induce DNA breaks in an in vitro model simulating the human intestinal barrier [106]. Moreover, GO nanosheets could break through the first line of host defense by disrupting the ultrastructure and biophysical properties of lung surfactant membranes [133]. Combined with the routes and doses of human exposure, relevant biological barriers toxicity can be considered as an aspect of assessing GFNs genotoxicity.

## 5. Genotoxicity Testing of GFNs

### 5.1. Detection of GFNs in Cells and Organism Tissues

The detection of GFNs internalization (distribution and behavior) in model organisms and cells is a key step for a better understanding of their genotoxicity and underlying mechanisms. The most commonly used detection technique includes direct observation of localization of GFNs in organisms and cells by transmission electron microscopy (TEM) [88]. The hyperspectral imaging is also used to visualize cellular interactions with NPs [134], such as cellular uptake and binding of GFNs [87]. The label-based approaches to image GFNs exist in cells by confocal and fluorescence microscopy, reflection-based imaging, and flow cytometry. Additionally, scanning electron microscopy (SEM) can be used to detect the attachment of GFNs in the surface zone of cells [52,87]. Raman spectroscopy and atomic force microscopy (AFM) were used to evaluate nuclear area changes and the disruption of DNA chains impacted by GQDs, respectively [69]. However, these traditional techniques are limited by low observation efficiency and large errors of quantitative results, with are disadvantages in the detection of GFNs [88]. Few studies focus on GFNs nuclear detecting techniques. In the biological imaging field, most research pays attention to safe application of fluorescent GFNs nuclear images rather than assessing genotoxicity of GFNs from an environmental toxicology point of view [135,136,137]. It is necessary to further optimize and develop detection techniques of GFNs in cells and organism tissues for a better understanding of genotoxicity. For example, Chen et al. [138] used laser desorption/ionization mass spectrometry imaging to map and quantify precisely the sub-organ distribution of the carbon nanotubes, GO, and carbon nanodots in mice. The SEM–Raman spectroscopy co-located system provide both SEM and Raman data from the same area on the cell sample, which avoids sample registration issues and makes observed results more accurate [139].

### 5.2. Genotoxicity Assay of GFNs

There are several assays available to access the genotoxicity of GFNs, measuring various endpoints [98]. The Ames test (bacterial reverse mutation), the comet assay (single cell gel electrophoresis), the chromosomal aberration (CHA), and micronuclei (MN) are the most common tests for genotoxicity. The Ames test (bacterial reverse mutation) can provide initial testing for genotoxicity. The comet assay can detect DNA damage, while the CHA and MN can test large chromosomal abnormalities. The hypoxanthine phosphoribosyl transferase (HPRT) gene is suitable for assessing mutations induced by suspect genotoxic agents, such as NPs [98]. Oxidative DNA damage should be considered one of the causes of genotoxicity. Superoxide radicals can lead to the activation of oxidation of the guanine bases present in the DNA strands, causing rupture to these strands. The most commonly used detection techniques include 8-hydroxydeoxyguanosine and 7, 8-dihydro-oxodeoxyguanine by HPLC with electrochemical detection [140].

## 6. Conclusions, Challenges, and Perspectives

On the basis of the existing literatures, we propose several genotoxic effects for GFNs in Figure 3. To date, there are few studies on genotoxicity mediated by direct interactions with DNA for GFNs (only GO and GQDs). That oxidative stress induced by GFNs causes DNA damage has been well established and studied. Regarding other indirect genotoxicity (e.g., epigenetic toxicity, inflammation, and autophagy), the studies largely focus on genotoxic effects induced by GFNs, and there is a lack of studies on the mechanisms underlying the observed effects. The genotoxicity of GFNs will depend on both inherent physicochemical properties (e.g., surface functionalization and coatings), exposure dose and times, and their fate in organisms or the environment. Although this review paper provides preliminary information on the genotoxicity of GFNs, the data is still very limited, especially with regard to the type of GFNs and exposure dose. The traditional techniques are limited by low observation efficiency and large errors of quantitative results, which are disadvantages in the detection of GFNs.

A number of issues remain in this area: (1) a lack of nuclear detecting and tracking techniques for GFNs to investigate the direct interactions of GFNs with DNA; (2) a challenge to reveal mechanisms underlying the indirect genotoxicity of GFNs, such as causal epigenetic mechanisms; and (3) an incomplete evaluation database regarding the type of GFNs, applied dosages, and exposure times, etc. These limitations are expected since genotoxicity research of NPs, especially GFNs, is still in their infancy when compared to other areas of toxicity (e.g., cytotoxicity, immunotoxicity, neurotoxicity, reproductive and developmental toxicity). Overall, further studies should address the questions mentioned above to clarify the genotoxic mechanisms of GFNs.

## Figures and Tables

**Figure 1 nanomaterials-11-02889-f001:**
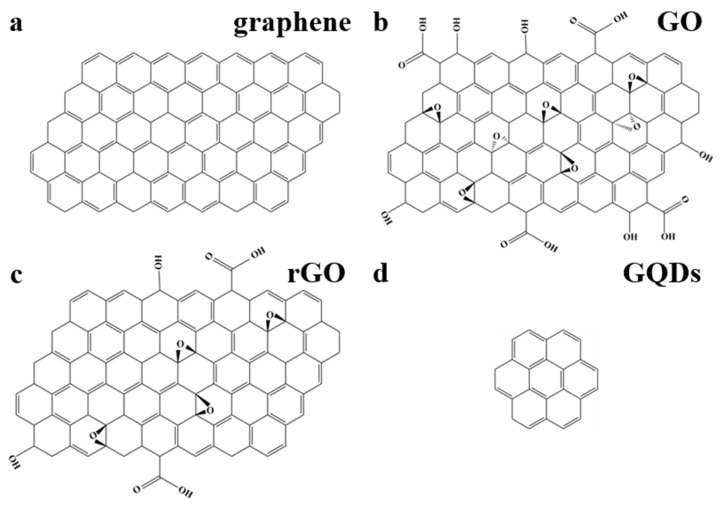
Structural models of single-layer graphene (**a**), graphene oxide (**b**), reduced graphene oxide (**c**), and graphene quantum dots (**d**).

**Figure 2 nanomaterials-11-02889-f002:**
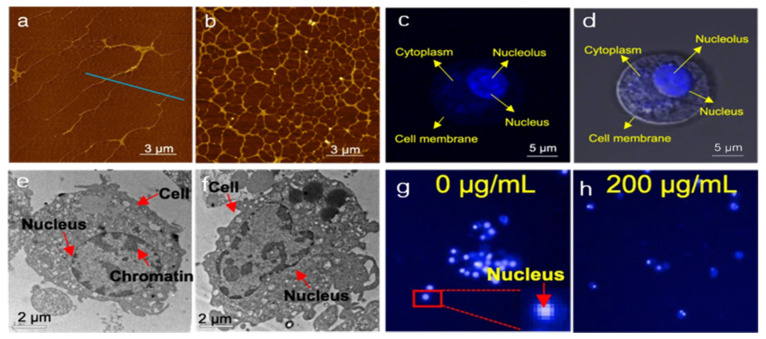
Effects of GQDs on the nucleus and DNA in the nucleus. (**a**,**b**) DNA chain damage caused by GQDs; (**c**,**d**) accumulation of GQDs; (**e**,**f**) nuclear damage by GQDs; (**g**,**h**) effect of GQDs on nuclear viability and area; (**a**,**c**,**e**,**g**) are the blank control groups, and (**b**,**d**,**f**,**h**) are the exposed groups of 200 mg/L GQDs for 24 h, reproduced from [69], from BioMed Central, 2018.

**Figure 3 nanomaterials-11-02889-f003:**
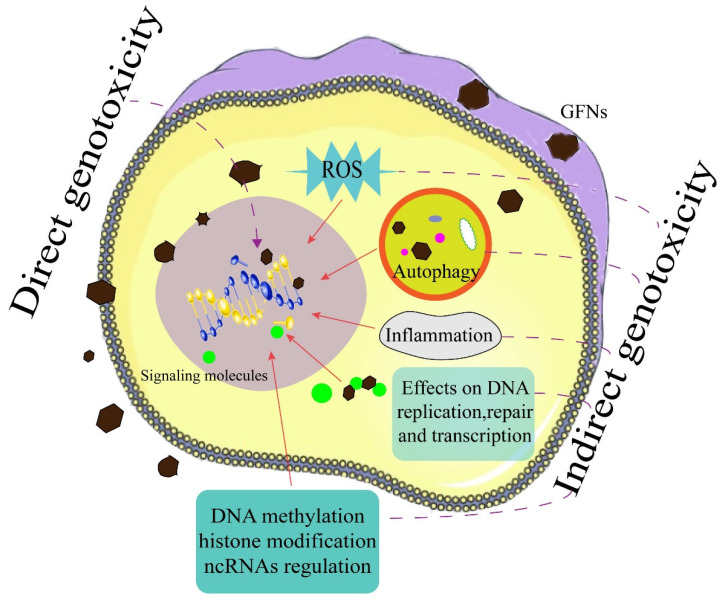
Direct and indirect effects of GFNs on DNA.

## Data Availability

Not applicable.

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
