# Peer review of "Direct and Indirect Genotoxicity of Graphene Family Nanomaterials on DNA—A Review"

_nanomaterials, 2021, doi:10.3390/nano11112889_

Round 1
Reviewer 1 Report
The paper provides an overview on the direct and indirect genotoxicity of graphene family nanomaterials. The topic is of great impact from the point of the potential toxicity and thus the regulatory aspects of nanomaterials. The paper is worth publishing after major revision.
- The abstract does not match the content of the paper. Given the vital role of genotoxicity, in conclusion, further steps are proposed without mentioning them in the manuscript (e.g., developing reliable testing methods; establishing multilevel dose indicators evaluation system).
- The wide variety of graphene-based products is outlined in the Introduction, but no products available on the market are named. It would be useful to add the product name to the Table 1.
- A separate subchapter is required about the testing method and regulatory approach of graphene-based products.
- It is known that a strong correlation was observed between particle size and cytotoxicity, demonstrating that the lower the particle size the higher the cytotoxicity, indicating size-dependent toxicity of nanomaterials. The surface characteristics, the attached surface ligands have also an impact on the biocompatibility of these materials.The surface functionalization of the silica particles affected the toxicity and the elimination. Furthermore, it was shown that the shape of these particles could modify the toxicity. Surface functionalization also have an impact of toxicity, it was demonstrated that AuNPs with cationic charge were more toxic than anionic particles [e.g. Szabó P. Current Pharmaceutical Design, 2015, 21, 3148-3157]. The authors should reflect the effect of physicochemical characteristics of the graphene-based products on their toxicity.
Author Response
Responses to Review 1’ Comments
Reviewer 1 Comments to Author
The paper provides an overview on the direct and indirect genotoxicity of graphene family nanomaterials. The topic is of great impact from the point of the potential toxicity and thus the regulatory aspects of nanomaterials. The paper is worth publishing after major revision.
Response: Thank you for all the comments. The responses to each comment are presented below.
- The abstract does not match the content of the paper. Given the vital role of genotoxicity, in conclusion, further steps are proposed without mentioning them in the manuscript (e.g., developing reliable testing methods; establishing multilevel dose indicators evaluation system).
Response: According to your comment, we revised the further steps in abstract and conclusion.
In abstract, to match the abstract with the content of the paper, the further step “2) establishing multilevel dose indicators evaluation system” has been changed to “3) enriching evaluation database regarding the type of GFNs, applied dosages and exposure times”. And the further step “3) elucidating the response mechanisms associated with genotoxicity in depth, especially the study of signaling pathways” has been changed to “2) elucidating the response mechanisms associated with genotoxicity in depth”. The above information has been clarified on page 1, lines 21-22 in the revised manuscript.
In conclusion, the question remains in this area “(1) a lack of nuclear detecting and tracking techniques for GFNs to investigate the direct interactions of GFNs with DNA” has been supported by “5.1. Detection of GFNs in Cells and Organism Tissues”. Furthermore, to further match the content of the paper, we have pointed out the inadequacy of research data regarding the type of GFNs, applied dosages and exposure times, etc. and the shortcoming of traditional detecting techniques of GFNs, as shown as on page11, lines 398-401 in the revised manuscript. The “(3) an incomplete evaluation system regarding the applied dosages and exposure times, etc.” has been changed to “(3) an incomplete evaluation database regarding the type of GFNs, applied dosages and exposure times”, and we have clarified this point on page 11, lines 406-407.
- The wide variety of graphene-based products is outlined in the Introduction, but no products available on the market are named. It would be useful to add the product name to the Table 1.
Response: Thank you for your advice. The market information of GFNs containing the commercial names of graphene-based products and the supplier or synthesis methods, has been added to Table 1, as shown on page 3, in line 76, in the revised manuscript.
- A separate subchapter is required about the testing method and regulatory approach of graphene-based products.
Response: We agree with your comment. We have added a separate subchapter on the “5. Genotoxicity Testing of GFNs”. To provide the testing method, the subchapter “5.1. Detection of GFNs in Cells and Organism Tissues” has been added, as shown on page 10, lines 353-375 in the revised manuscript. To provide the regulatory approach about the genotoxicity of GFNs, the subchapter “5.2. Genotoxicity Assay of GFNs” has been added, as shown on page 11, lines 376-388 in the revised manuscript.
- It is known that a strong correlation was observed between particle size and cytotoxicity, demonstrating that the lower the particle size the higher the cytotoxicity, indicating size-dependent toxicity of nanomaterials. The surface characteristics, the attached surface ligands have also an impact on the biocompatibility of these materials. The surface functionalization of the silica particles affected the toxicity and the elimination. Furthermore, it was shown that the shape of these particles could modify the toxicity. Surface functionalization also have an impact of toxicity, it was demonstrated that AuNPs with cationic charge were more toxic than anionic particles [e.g. Szabó P. Current Pharmaceutical Design, 2015, 21, 3148-3157]. The authors should reflect the effect of physicochemical characteristics of the graphene-based products on their toxicity.
Response: We would like to thank you for the comments. The factors affecting genotoxicity are similar to those affecting cytotoxicity for GFNs, thus, we have briefly described the factors influencing the cytotoxicity of GFNs, as shown as on page 7, lines 251-254 in the revised manuscript [Szabo, P., 2015].
Furthermore, we have added relevant examples to describe the effects of functionalization of oxygen-containing and other functional groups, surface charge and hydrophilicity on the genotoxicity of GFNs, as shown on page 7, lines 267-267 and page 8, lines 268-270, lines 278-301 in the revised manuscript [Krasteva, N., 2019; Hashemi, E., 2014; Sasidharan, A., 2011].
Reference:
- Szabo, P.; Zelko, R., Formulation and stability aspects of nanosized solid drug delivery systems. Curr. Pharm. Des. 2015, 21 (22), 3148-3157.
- Krasteva, N.; Keremidarska-Markova, M.; Hristova-Panusheva, K.; Andreeva, T.; Speranza, G.; Wang, DY.; Draganova-Filipova, M.; Miloshev, G.; Georgieva, M., Aminated graphene oxide as a potential new therapy for colorectal cancer. Oxid. Med. Cell. Longevity. 2019, 2019, 3738980
- Hashemi, E.; Akhavan, O.; Shamsara, M.; Rahighi, R.; Esfandiar, A.; Tayefeh, AR., Cyto and genotoxicities of graphene oxide and reduced graphene oxide sheets on spermatozoa. RSC Adv. 2014, 4 (52), 27213-27223.
- Sasidharan, A.; Panchakarla, LS.; Chandran, P.; Menon, D.; Nair, S,; Rao, CNR.; Koyakutty, M., Differential nano-bio interactions and toxicity effects of pristine versus functionalized graphene. Nanoscale . 2011, 3 (6), 2461-2464.

Reviewer 2 Report
See atachment.

Author Response
Responses to Review 2’ Comments
Reviewer 2 Comments to Author
The paper Direct and Indirect Genotoxicity of Graphene Family Nanomaterials on DNA A Review presents a literature survey concerning graphene family nanomaterials that includes graphene, graphene oxide, reduced graphene oxide and graphene quantum dots, respectively and more important establish some future aspects that must be clarified in the field. The topic is actual and important considering that the influence of this class of innovative nanomaterials on medium and organisms is poorly known. All data are clear presented and sustained by mechanism of action and factors associated with these materials genotoxicity.
As result recommend minor corrections such as follows:
Response: Thank you for all the comments. The responses to each comment are presented below.
- biological medicine must be replaced by medicine
Response: The words “biological medicine” have been replaced by “medicine”, as shown on page 1, line 38 in the revised manuscript.
- in vivo and intro must be replaced by in vivo and in vitro (and the Italic style must be verified for these in whole paper)
Response: Thank you for your advices. The words “in vivo and intro” have been replaced by “in vivo and in vitro” and the Italic style has also been verified, as shown on page 1, line 43, page 3, line 76, and page 5, line 150 in the revised manuscript.
- DNA Methylation must be provided as DNA methylation
Response: The words “DNA Methylation” have been changed to “DNA methylation”, as shown on page 6, lines 179-180 in the revised manuscript.
- In the expression “amino, chromatin and DNA” the type of amino derivative must be specified (amino acids?)
Response: The expression “amino, chromatin and DNA” has been changed to “nuclear lamina, chromatin and DNA”, as shown on page 7, line 235 in the revised manuscript.
- The expression “and Autophagy may” must be corrected as “and autophagy may”
Response: The expression “and Autophagy may” has been corrected as “and autophagy may”, as shown on page 7, line 241 in the revised manuscript.

Round 2
Reviewer 1 Report
The authors adequately addressed the referee's comments and modified the paper accordingly. Therefore, I suggest the publication of the revised version.